# Comparison Between Transient Elastography and Point Shear Wave Elastography in the Assessment of Liver Fibrosis According to the Grade of Liver Steatosis

**DOI:** 10.3390/jcm14155417

**Published:** 2025-08-01

**Authors:** Giuseppe Losurdo, Antonino Castellaneta, Claudia Di Nuccio, Paola Dell’Aquila, Ilaria Ditonno, Domenico Novielli, Antonio Continisio, Margherita De Bellis, Alfredo Di Leo, Mariabeatrice Principi, Michele Barone

**Affiliations:** Section of Gastroenterology, Department of Precision and Regenerative Medicine and Jonian Area, University of Bari, Piazza Giulio Cesare 11, 70124 Bari, Italy

**Keywords:** liver steatosis, fibrosis, stiffness, Fibroscan^®^, shear wave elastography

## Abstract

**Background**: Transient elastography (TE), using Fibroscan^®^ and point shear wave elastography (pSWE), are two techniques used to estimate liver fibrosis. The aim of our study was to compare, for the first time, these two techniques in Metabolic Dysfunction-Associated Steatotic Liver Disease (MASLD), stratifying the analysis on the basis of the grades of steatosis. **Methods**: We recruited 85 consecutive MAFLD patients who underwent liver stiffness (LS) measurement performed by Fibroscan^®^ and pSWE on the same day. Severity of steatosis was estimated by Fibroscan^®^ and expressed as controlled attenuation parameter (CAP), ranging from S0 to S3. Spearman’s “r” coefficient was used to calculate the correlation and Bland–Altman graphs was used to evaluate the agreement. **Results**: In general, the correlation and agreement between Fibroscan^®^ and pSWE were substantial (r = 0.66, *p* < 0.001 and bias= −0.64 ± 2.48, respectively). When data were analyzed according to the grade of steatosis, an increasing significant correlation was observed going from S0 to S2 (r = 0.79, r = 0.81, and r = 0.85, respectively), whereas a low correlation and agreement were observed for S3 patients (r = 0.48, *p* = 0.003, bias= −0.95 ± 2.51). **Conclusions**: Fibroscan^®^ and pSWE are equivalent techniques to estimate liver fibrosis in patients with mild to moderate steatosis, while in presence of severe steatosis their agreement is low.

## 1. Introduction

Metabolic dysfunction-associated steatotic liver disease (MASLD) is a condition characterized by ectopic accumulation of lipids in hepatocytes associated with a metabolic dysfunction and in the absence of daily alcohol consumption greater than or equal to 20 g in women and 30 g in men, as well as other secondary causes of lipid accumulation in the liver [1]. MASLD covers a spectrum of clinical-pathological conditions ranging from simple steatosis, the so-called steatoitic liver disease (SLD), to metabolic dysfunction-associated steatohepatitis (MASH), passing through different stages of fibrosis and progressing to hepatic cirrhosis. Patients with MASH are characterized by hepatic steatosis, ballooning degeneration of hepatocytes and lobular inflammation. This inflammatory process not only increases the risk of progression towards cirrhosis, but several studies have also shown that patients with MASH are more predisposed to develop hepatocellular carcinoma compared to the general population [2], even in absence of cirrhosis [3].

The diagnosis of MASLD is based on a multidisciplinary approach that includes medical history, clinical examination, laboratory tests and imaging techniques, in addition to liver biopsy that remains the gold standard for diagnosis [4,5]. Nevertheless, liver biopsy is no longer routinely indicated as it is an invasive technique with potential harmful effects and limitations due to sampling and observational bias of the sample tissue. As a result, among the alternative non-invasive methods for liver fibrosis assessment, transient elastography (TE) by Fibroscan^®^ and shear wave elastography (pSWE) have a large consensus.

TE is performed by a Fibroscan^®^ device (EchoSens, Paris, France), which consists of an ultrasound transducer probe mounted on the axis of a vibrating system. The emission of low-frequency vibrations at the level of the right intercostal spaces generates a shear wave that propagates through the liver. The propagation speed of this wave is detected using a pulse-echo ultrasound acquisition system, considering that the speed is directly proportional to tissue stiffness (expressed in kPa). Compared to pSWE, Fibroscan^®^ has the advantage of detecting hepatic steatosis in patients with MASLD/MASH, even when hepatocellular fat accumulation is around 10%, without being influenced by fibrosis or cirrhosis [6]. Fibroscan estimates fat accumulation through the controlled attenuation parameter (CAP) [6].

Point shear wave elastography (pSWE) is a more recent non-invasive procedure useful for detecting and staging liver fibrosis, with a significant correlation observed between the AST to Platelet Ratio Index (APRI) score and the pSWE score [7]. pSWE is based on the Acoustic Radiation Force Impulse (ARFI) technique: liver fibrosis is assessed by targeting a specific region of interest through the right intercostal spaces and this region’s elastic properties are analyzed using a region of interest (ROI) cursor during real-time B-mode ultrasound imaging [6]. In this regard, pSWE diverges from TE as it allows liver evaluation through B-mode ultrasound imaging control, which aids in screening and diagnosis of liver cirrhosis or focal liver lesions and helps to avoiding off-target measurements—both intrahepatic (for instance bile ducts and blood vessels) and extrahepatic (e.g., kidney, ribs, etc.), by focusing on the best acoustic window [8].

Since fibrosis is frequently associated with steatosis, especially in the MASLD. The aim of our preliminary study was to compare, for the first time, the assessment of liver fibrosis by these two techniques in the presence of different grades of steatosis.

## 2. Materials and Methods

### 2.1. Patients Enrollment

This prospective study was conducted in 85 consecutive patients with SLD undergoing Fibroscan^®^ and pSWE the ultrasound service of Gastroenterology Unit of Policlinic University Hospital, Bari, Italy. The study was conducted in accordance with the Declaration of Helsinki and was approved by the local Ethic Committee (Prot. No. 0085420/06/10/2022). All patients enrolled in the study signed informed consent forms, agreeing to the use of their data for research purposes.

Patients were included if they had a clinical diagnosis of SLD and an ultrasound picture suggestive of steatosis according to the ultrasound classification by Paige and elsewhere [9,10]. Patients with SLD of both sexes, without ethnic distinction and aged 18 years or older were enrolled. To improve the accuracy and reliability of the measurements we excluded patients with central obesity (defined by abdominal circumference > 102 cm for men and >88 cm for women), perihepatic ascites, acute hepatitis with elevated transaminases (AST or ALT greater than 5 times the upper limit of normality), right heart failure, very narrow intercostal spaces, a history of liver transplantation or thoracic surgery involving manipulation of the rib cage. The condition of normal weight or obesity was assessed by measuring the body mass index (BMI), expressed as body weight kg/height (m^2^). Patients deemed unsuitable for liver fibrosis measurement by the physician were also excluded from the study.

For each enrolled patient the following data were collected: demographic data (sex and age), liver stiffness assessed by Fibroscan^®^ and pSWE corresponding F score and CAP [steatosis] with corresponding S score); liver assessment data using pSWE (fibrosis with corresponding F score); laboratory data including ALT, AST, GGT, ALP serum levels, (expressed as a ratio relative to the upper limit of normal), total bilirubin (n.v. < 1 mg/dL), and platelet count (n.v > 150.000), and clinical data (BMI, diabetes and hypertension).

### 2.2. TE Assessment

Liver fibrosis measurements by Fibroscan^®^, using a M size probe (EchoSens, Paris, France), and pSWE (QElaXto program on Esaote MyLab Xpro80, Genoa, Italy) were assessed consecutively on the same day. Both techniques were performed by placing the probe over the right liver lobe through the intercostal spaces, with the patient in a supine position and the right arm in maximum abduction. Patients were instructed to hold their breath during data acquisition and fasting was required on the day of the examination [5].

During TE via Fibroscan^®^ the tip of the transducer was covered with coupling gel and positioned between the ribs over the right liver lobe. The operator, guided by A-mode ultrasound images provided by the system, identified a liver region at least 6 cm thick and free from large vascular structures, then initiated data acquisition by pressing the button on the probe. The software automatically discarded acquisitions that did not show a correct vibration pattern or a proper propagation follow-up [6].

Similarly, during liver fibrosis measurement with pSWE patients were asked to hold their breath during data acquisition, avoiding deep inspiration before apnea [6]. The region of interest was identified 1 to 3 cm under the liver capsule, avoiding vessels and bile ducts, with the probe positioned in the intercostal spaces intersecting the right axillary line. During measurements, minimal pressure was applied to avoid falsely increased stiffness values [11,12]. In both procedures liver fibrosis was measured in kPa and at least 10 measurements were acquired to ensure reliable data.

Staging of fibrosis was classified according to Metavir classification [13,14]. Cut-offs for the different stage of fibrosis were validated according to the 43rd Annual Congress of the European Association for the Study of the Liver (EASL). Specifically, liver fibrosis values < 7, ≥7.0 kPa, 8.7 kPa, 10.3 kPa and 14 kPa were regarded as indicative of absent (F0), mild (F1), moderate (F2), advanced (F3) fibrosis and cirrhosis (F4), respectively [13,14].

Cut-offs provided by Fibroscan^®^ manufacturer were used to stage liver steatosis: CAP values < 248 dB/m, ≥248 dB/m, 268 dB/m and 280 dB/m are recommended for staging absent (S0), mild (S1), moderate (S2) and severe (S3) steatosis, respectively.

### 2.3. Statistical Analysis

Continuous data were expressed as means and standard deviations and Student’s T-test or ANOVA were used when comparing two or more groups, respectively. Dichotomous data were expressed as proportions or percentages and were compared by chi-square test. The correlation between continuous variables was assessed by calculating Spearman’s correlation coefficient r (with 95% CI). The strength of correlation was defined as perfect if r = 1, very strong if 0.8 < r < 1, strong if 0.6 < r < 0.8 and moderate if 0.4 < r < 0.6. Bland–Altman plots were generated to assess the agreement between continuous data series by calculating the bias and its corresponding 95% CI: if the bias fell within the mean’s confidence interval, the two methods were considered interchangeable.

A multivariate analysis by logistic regression was conducted to evaluate factors associated with the severity of steatosis and significant factors were expressed as odds ratios (OR) with the corresponding 95% CI. Linear logistic regression was also performed to assess factors correlated with the CAP value and significant factors were expressed as b coefficients with the corresponding 95% CI.

All statistical analyses were performed two-tailed, with limit of significance set at *p* < 0.05 and were performed using GraphPad Prism version 5.0 and SPSS version 23 (IBM).

## 3. Results

### 3.1. Patients’ Characteristics

A total of 85 patients with steatotic liver were recruited [22 women (25.9%) and 63 men (74.1%)], with a mean age of 54.7 ± 13.1 years, for this preliminary study. Table 1 shows patients’ demographic, clinical, laboratory data, and values of stiffness assessed by Fibroscan^®^ and pSWE. The comorbidities analyzed included diabetes, hypertension and obesity. The 21.9% of the patients were diabetic and 62.5% suffered from arterial hypertension. Regarding BMI, a mean value of 28.6 ± 3.5 was observed.

### 3.2. Procedural Parameters

Fibroscan^®^ assessment demonstrated the absence of liver stiffness (F0) in 51 patients (60%), 16 patients (18.8%) were stage F1, 4 patients (4.7%) stage F2, five patients (5.9%) stage F3 and 9 patients (10.6%) stage F4. Liver steatosis was also assessed, resulting S0 in 25 patients (29.4%), S1 in 11 patients (12.9%), S2 in 13 patients (15.3%), and S3 in 36 patients (42.4%).

Using pSWE 58 patients (68.2%) were identified as having fibrosis stage F0, 13 patients (15.3%) as F1, 2 patients (2.4%) as F2, 5 patients (5.9%) as F3, and 7 patients (8.2%) as F4. All these data are summarized in Table 2.

### 3.3. Correlation and Agreement Between pSWE and Fibroscan^®^

The correlation between LS measured by pSWE and Fibroscan^®^ was assessed by calculating Spearman’s correlation coefficient (r) with a 95% confidence interval (CI). Considering all patients, the correlation was good, with an r value of 0.66 (0.52–0.77 CI) and *p* < 0.001 (Figure 1a).

The agreement between the two techniques was evaluated by Bland–Altman plots Considering all patients a bias = −0.64 ± 2.48 was observed, with 5 values falling outside the CI of the mean (Figure 1b).

In the subcategory of patients with absent steatosis, the Spearman’s correlation coefficient (r) value was high (0.79; 0.57–0.91 CI) with a *p* < 0.001, indicating a substantial correlation between the two techniques (Figure 2a). In the same patient category (S0) a Bland–Altman plot showed a bias of −0.58 ± 2.72, with only 2 values falling outside the confidence interval of the mean (Figure 2b).

In the category of patients with mild steatosis (S1) (Figure 3a) a significant correlation was observed with r = 0.81 (0.39–0.95% CI; *p* = 0.002), and a bias of 0.19 ± 3.11 was calculated, with only one value falling outside the confidence interval of the mean as shown in Figure 3b.

In the subgroup of patients with moderate steatosis (S2) Spearman’s coefficient had a value of r = 0.85 (0.56–0.96 CI) with *p* < 0.001, demonstrating a substantial correlation between LS measured by Fibroscan^®^ and LS measured by pSWE (Figure 4a). In the same subgroup of patients (S2) a bias of −0.60 ± 1.07 was calculated, with only one value falling outside the confidence interval of the mean (Figure 4b).

In the last subgroup of patients, those with severe steatosis (S3), a lower correlation between the two techniques was observed compared to the previous subgroups. In fact, a correlation coefficient r = 0.48 (0.17–0.70 CI) with *p* = 0.003 was calculated, as shown in Figure 5a and the bias was of −0.95 ± 2.51, with only 2 values falling outside the confidence interval of the mean (Figure 5b).

### 3.4. Associated Factors

A multivariate logistic regression analysis was conducted to investigate factors associated with the severity of hepatic steatosis. Significant factors were expressed as odds ratios with corresponding 95% confidence intervals: the only significant factor was diabetes, with an OR = 348.7 (95% CI: 322–365) and *p* < 0.001, as shown in Table 3.

To evaluate factors associated with the CAP value a linear regression analysis was performed and significant factors were expressed as coefficients b with corresponding 95% confidence intervals. We found an association between CAP value and BMI, with a coefficient b = 0.47 (*p* = 0,052).

## 4. Discussion

The exploration of liver fibrosis is essential in the management of patients with MASLD as it allows us to choose the most appropriate therapeutic approach and to stratify each patient based on the risk of complications such as cirrhosis and hepatocellular carcinoma [2,3]. Liver biopsy is the gold standard to stage fibrosis, but it is an invasive procedure with possible adverse events, thus non-invasive methods such as TE via Fibroscan^®^ and pSWE are gaining growing consent.

In a study conducted on 349 patients with chronic liver diseases with heterogeneous etiology, the two elastographic techniques were compared with liver biopsy, showing a correlation coefficient r = 0.79 for SWE and 0.70 for TE, thus indicating a significant correlation [15]. Moreover, the AUROC (areas under the ROC curve) of SWE and TE were 0.89 and 0.86, respectively, for the diagnosis of mild fibrosis; 0.88 and 0.84 for significant fibrosis; 0.93 and 0.87 for severe fibrosis; 0.93 and 0.90 for the diagnosis of cirrhosis. In two other multicenter studies [16,17] correlation coefficients of 0.88 and 0.867 were reported, respectively, when comparing TE with pSWE.

These are some of the studies that have confirmed the accuracy and agreement of these two elastographic techniques in the assessment of liver fibrosis. In fact, the European Federation of Societies for Ultrasound in Medicine and Biology (EFSUMB) has recommended TE, pSWE, and 2D-SWE as first-line techniques for staging liver fibrosis and ruling out cirrhotic progression in patients with MASLD [18]. The European Association for the Study of the Liver (EASL) has also emphasized that pSWE shows an equivalent diagnostic performance compared to TE in diagnosing advanced stages of fibrosis or cirrhosis, while it is less sensitive in intermediate stages of fibrosis. This is consistent with several studies [13,19,20] that have compared TE and pSWE across different fibrosis stages, observing that the AUROC of the two techniques were similar in advanced fibrosis, whereas in intermediate stages pSWE showed a smaller AUROC (i.e., lower accuracy).

Unlike these previous studies that compared the two techniques based on the degree of fibrosis, in the present study we analyzed the correlation and agreement between TE via Fibroscan^®^ and pSWE according to different grades of steatosis. Our results showed that in patients with absent, mild or moderate steatosis there was a substantial correlation between the two techniques, with r = 0.79 in S0 patients (strong correlation), 0.81 in S1 patients and 0.85 in S2 patients (very strong correlation). However, these high values were not confirmed in patients with severe steatosis (S3), where we calculated a correlation coefficient r = 0.48 (moderate correlation). This value clearly influenced the overall correlation value across the entire patient cohort, resulting in a coefficient r = 0.66 (strong correlation).

This lower correlation between LS measured by TE and LS measured by pSWE in patients with severe steatosis (S3) was also reflected in the bias values obtained using Bland–Altman plots. As previously mentioned, these plots allowed us to assess the agreement between the two techniques, considering the methods interchangeable when the bias fell within the confidence interval of the mean. While in S0, S1 and S2 patients the bias remained low (respectively, −0.58; 0.19; −0.60), it increased in patients with S3 steatosis, with a calculated value of −0.95 ± 2.51.

Therefore, the presence of severe hepatic steatosis may interfere with the propagation of elastic waves through the liver, thereby affecting the stiffness measurement. Specifically, this great lipid component is likely to alter the transmission of vibrations, making it more difficult to obtain an accurate evaluation of liver stiffness. As a result, stiffness measurements are influenced not only by liver fibrosis, but also by the amount of fat present, reducing the accuracy of both TE and pSWE in patients with severe steatosis. In one study [21], in fact, it was observed that among patients without significant fibrosis (F0–F1) as well as those without severe fibrosis (F0–F2), falsely elevated LS values were more frequently measured in individuals with ≥66% steatosis compared to those without (F0–F1: 6.9 vs. 5.8, *p* = 0.04; F0–F2: 7.4 vs. 6.0, *p* = 0.001). These findings led to the conclusion that, in MASLD patients, severe steatosis should always be carefully assessed in order to avoid overestimation of liver fibrosis when using TE.

Another study [22] reported similar results by comparing LS values measured by MRE (magnetic resonance elastography) and 2D-SWE. The results indicated that severe hepatic steatosis could lead to an overestimation of stiffness measured by 2D-SWE in patients with no or mild fibrosis, thereby reducing its specificity in detection.

MRE is therefore another non-invasive technique used to assess liver fibrosis and in doing so it has proven to be more accurate than other elastographic techniques, but it is a high-cost technique with long execution times, and it is also limited by patient cooperation, patient-related characteristics and its inability to classify liver fibrosis based on different etiologies [23].

Our study has some limitations: first of all, the small sample size and, consequently, the limited number of TE and pSWE procedures performed. However, despite the limited sample size, using multivariate logistic regression to assess factors associated with the severity of steatosis, we were able to calculate an odds ratio (OR) of 348.7 for diabetes (95% CI 322–365; *p* < 0.001). This result is not surprising considering diabetes is one of the main risk factors for MASLD [24].

Linear logistic regression was used to evaluate factors associated with the CAP value, among which BMI was particularly significant as demonstrated by a coefficient b = 0.47 (*p* = 0.052), indicating a direct relationship whereby increasing BMI corresponded to higher CAP values. These results confirm what other studies [25,26] have highlighted, confirming a positive relationship between these two parameters.

Estimation of fibrosis by pSWE has some drawbacks. Being a relatively novel techniques, some authors advocate different cut-off to distinguish the different fibrosis interval [27]. Nevertheless, we decided to use the same cut-offs both for TE and pSWE.

In this regard, another limitation of our study is the exclusion of patients with severe obesity. This decision was made due to the unavailability of an XL probe for the Fibroscan^®^ device.

Finally, an additional limitation of our study is the absence of confirmatory biopsies for the degrees of fibrosis and steatosis detected by TE via Fibroscan^®^ and pSWE. However, the aim of our study was solely to compare the two non-invasive techniques and assess their agreement, not to evaluate their accuracy against the invasive biopsy procedure.

## 5. Conclusions

In conclusion, our preliminary study highlights a substantial correlation between transient elastography via Fibroscan^®^ and point shear wave elastography in the assessment of liver fibrosis. This correlation is particularly strong in cases of absent or intermediate steatosis (S1–S2), while it significantly decreases in the presence of severe steatosis (S3).

## Figures and Tables

**Figure 1 jcm-14-05417-f001:**
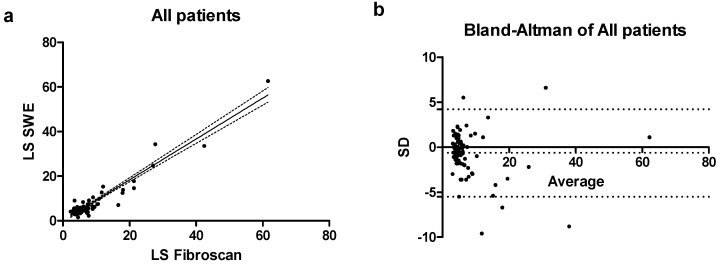
(**a**) Scatterplot reporting the correlation between Fibroscan^®^ and pSWE in the entire patient’s cohort; (**b**) Bland–Altman plot in the entire patients’ cohort.

**Figure 2 jcm-14-05417-f002:**
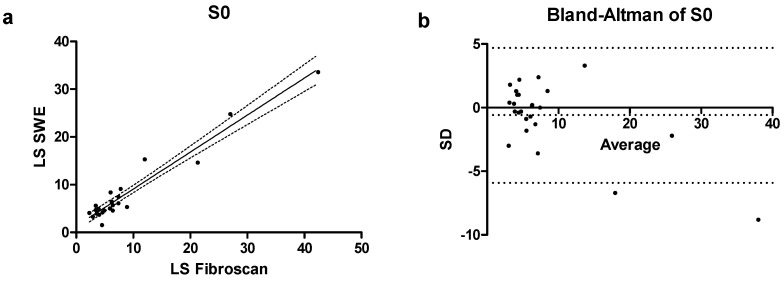
(**a**) Correlation scatterplot and (**b**) Bland–Altman plot of S0 patients enrolled in the study.

**Figure 3 jcm-14-05417-f003:**
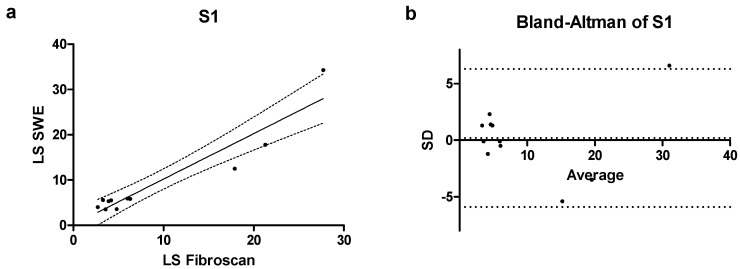
(**a**) Correlation scatterplot and (**b**) Bland–Altman plot of S1 patients enrolled in the study.

**Figure 4 jcm-14-05417-f004:**
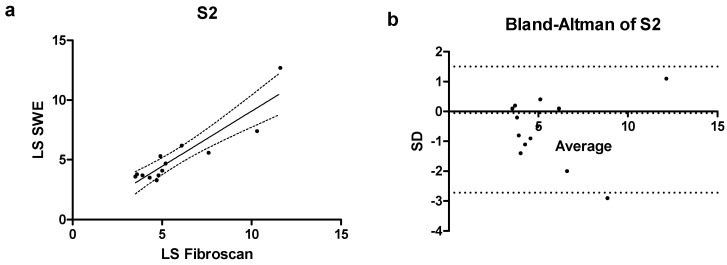
(**a**) Correlation scatterplot and (**b**) Bland–Altman plot of S2 patients enrolled in the study.

**Figure 5 jcm-14-05417-f005:**
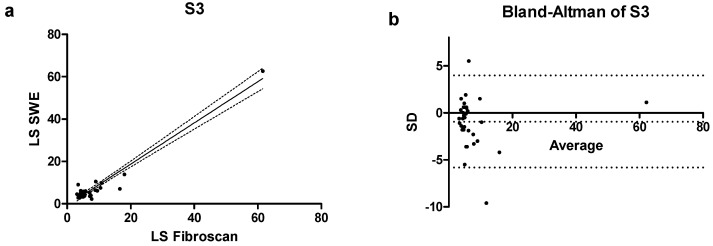
(**a**) Correlation scatterplot and (**b**) Bland–Altman plot of S3 patients enrolled in the study.

**Table 1 jcm-14-05417-t001:** Demographic, clinical and procedural characteristics of patients enrolled in the study.

Variables	N (%) o Mean ± SD
Age	54.72 ± 13.15
Sex	F 22 (25.9%) M63 (74.1%)
Diabetes	7 (21.9%)
Hypertension	20 (62.5%)
BMI	28.64 ± 3.54
ALT	0.87 ± 0.45
AST	0.85 ± 0.34
Total Bilirubin	0.86 ± 0.48
GGT	2.11 ± 5.80
ALP	0.74 ± 0.18
PLT	204.28 ± 60.07
LS Fibroscan^®^	7.97 ± 8.61
CAP	271.04 ± 52.78
LS pSWE	7.33 ± 8.24

BMI: body mass index; ALT: alanine aminotransferase; AST: aspartate aminotransferase; GGT: gamma-glutamyl transferase; ALP: alkaline phosphatase; PLT: platelets; LS: liver stiffness; CAP: controlled attenuation parameter.

**Table 2 jcm-14-05417-t002:** Procedural parameters of patients enrolled in the study.

Variables	N (%)
F0 Fibroscan^®^	51 (60%)
F1 Fibroscan^®^	16 (18.8%)
F2 Fibroscan^®^	4 (4.7%)
F3 Fibroscan^®^	5 (5.9%)
F4 Fibroscan^®^	9 (10.6%)
S0	25 (29.4%)
S1	11 (12.9%)
S2	13 (15.3%)
S3	36 (42.4%)
F0 pSWE	58 (68.2%)
F1 pSWE	13 (15.3%)
F2 pSWE	2 (2.4%)
F3 pSWE	5 (5.9%)
F4 pSWE	7 (8.2%)

**Table 3 jcm-14-05417-t003:** Multivariate analysis of factors associated with severity of steatosis.

Variable	OR (95% CI)	*p*
ALT	0.98 (0.0006–968)	1
AST	0.48 (0.0012–689)	0.98
GGT	4.8 (0.044–21.2)	0.78
PLT	3.14 (0.003–87.7)	0.97
BMI	732 (0.004–11,569)	0.92
Diabetes	348.7 (332–365)	<0.001
Hypertension	0.83 (0.16–4.44)	0.83

## Data Availability

The original contributions presented in this study are included in the article. Further inquiries can be directed to the corresponding author.

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
