# Peer review of "Comparison Between Transient Elastography and Point Shear Wave Elastography in the Assessment of Liver Fibrosis According to the Grade of Liver Steatosis"

_jcm, 2025, doi:10.3390/jcm14155417_

Round 1
Reviewer 1 Report
Comments and Suggestions for Authors
Dear Authors,
Thank you for this interesting topic you have chosen.
I appreciate the fact that in the Introduction part you have explained the methods you were going to reffer to and the context.
However, in the part dedicated to Material and methods, you explained very clearly the cut-off for Fibroscan, but not for pSWE, which it is still a hot topic.
Please, mention what you consider "central obesity".
Please, comment the results of the coefficients calculated and the p value.
I am looking forward to hearing from you.
Author Response
Dear Authors,
Thank you for this interesting topic you have chosen.
I appreciate the fact that in the Introduction part you have explained the methods you were going to reffer to and the context.
We thank the reviewer for the kind appreciation of our article.
However, in the part dedicated to Material and methods, you explained very clearly the cut-off for Fibroscan, but not for pSWE, which it is still a hot topic.
For our analysis, we used the same cut-off for both Fibroscan and SWE. We are aware that standardized cut-offs for SWE have not been established and that some authors advocate different ranges (Ronot M, Ferraioli G, Müller HP, Friedrich-Rust M, Filice C, Vilgrain V, Cosgrove D, Lim AK. Comparison of liver stiffness measurements by a 2D-shear wave technique and transient elastography: results from a European prospective multi-centre study. Eur Radiol. 2021 Mar;31(3):1578-1587). We discussed this point in the revised manuscript.
Please, mention what you consider "central obesity".
Central obesity was defined by abdominal circumference >102 cm for men and >88 cm for women.
Please, comment the results of the coefficients calculated and the p value.
In the Discussion, we added the entity of correlation, which is very strong if 0.8<r<1, strong if 0.6<r<0.8 and moderate if 0.4<r<0.6. In the case of r coefficients, the p value is only related to the entity of the sample size, therefore its significance is negligeable (see Barone M et al, Nutrition 2022).
I am looking forward to hearing from you.
Reviewer 2 Report
Comments and Suggestions for Authors
The Authors have presented an interesting study on comparision of two imaging methods for evaluation of liver fibrosis in MAFLD patients. Although the topic is current and potentially useful to readers, several important comments must be addressed.
1. I suggest to change the title of the article. "Liver stiffness" should be replecaed with "liver fibrosis", because this is the phenomenon actually assessed by the liver stiffness (LS) parameter. When referring to the METAVIR scale (F0-F4) in the description of the results, the term "fibrosis" and not "stiffness" should also be used. Moreover, the article should be considered a "preliminary study" due to the small number of subjects.
2. In the introduction, the Authors should describe fatty liver disease as MASLD, instead of MAFLD, according to the current Delphi consensus. Therefore, the number of subjects meeting MASLD criteria should be verified. These criteria should be clearly described in the methodology.
3. The authors should describe the inclusion and exclusion criteria for the study more clearly, providing cut-off points for some variables (central obesity, ALT and AST activity etc.). Please use the term "right ventricular failure" instead of "right heart failure".
4. Figures can be improved - please provide high-resolution images.
5. Point 3.4 in the results section can be omitted- the analysis of factors associated with severity of steatosis is not related to the topic of the article.
Author Response
The Authors have presented an interesting study on comparision of two imaging methods for evaluation of liver fibrosis in MAFLD patients. Although the topic is current and potentially useful to readers, several important comments must be addressed.
- I suggest to change the title of the article. "Liver stiffness" should be replecaed with "liver fibrosis", because this is the phenomenon actually assessed by the liver stiffness (LS) parameter. When referring to the METAVIR scale (F0-F4) in the description of the results, the term "fibrosis" and not "stiffness" should also be used. Moreover, the article should be considered a "preliminary study" due to the small number of subjects.
We agree with the comment of the reviewer. Therefore, when indicated, fibrosis replaced stiffness and the fact that this was a preliminary study was underlined repeatedly in the modified text.
- In the introduction, the Authors should describe fatty liver disease as MASLD, instead of MAFLD, according to the current Delphi consensus. Therefore, the number of subjects meeting MASLD criteria should be verified. These criteria should be clearly described in the methodology.
We agree with the comment of the reviewer, therefore, in the Introduction we replaced MAFLD with MASLD, according to new consensus. According to such consensus, we recruited patients with steatotic liver disease (SLD) (n = 85), of which 34 had at least one metabolic disease, based on which MASLD could be diagnosed. Considering the aim of our study, i.e. ultrasound concordance between two techniques, we deemed to include all SLD patients.
- The authors should describe the inclusion and exclusion criteria for the study more clearly, providing cut-off points for some variables (central obesity, ALT and AST activity etc.). Please use the term "right ventricular failure" instead of "right heart failure".
We performed the requested corrections.
- Figures can be improved - please provide high-resolution images.
We added figures with better definition.
- Point 3.4 in the results section can be omitted- the analysis of factors associated with severity of steatosis is not related to the topic of the article.
In this case, we respectfully do not agree with the comment of the reviewer. As we believe that this is an noteworthy analysis that may be of interest for the journal’s audience, we decided to leave it in the revised text.
Round 2
Reviewer 2 Report
Comments and Suggestions for Authors
The Authors have made sufficient changes in the manuscript. The paper can be published in present form.